Spatial and temporal patterns of human Puumala virus (PUUV) infections in Germany

Cunze Sarah cunze@bio.uni-frankfurt.de sarahcunze@gmail.com 1 2
Kochmann Judith 1 2
Kuhn Thomas 1 2
Frank Raphael 3
Dörge Dorian D. 1 2
Klimpel Sven 1 2
1 Goethe University Frankfurt, Institute of Ecology, Diversity and Evolution , Frankfurt am Main , Germany
2 Senckenberg Biodiversity and Climate Research Centre, Senckenberg Research Institute and Natural History Museum , Frankfurt am Main , Germany
3 Institute of Medical Microbiology and Hospital Hygiene, Heinrich-Heine Universität Düsseldorf , Düsseldorf , Germany
Ewer Katie
Electronic publication date: 2018 Feb 1
Publication date: 2018
Volume: 6
Electronic Location ID: e4255
Received 2017 Sep 11; Accepted 2017 Dec 19
Copyright: ©2018 Cunze et al.
Copyright year: 2018
Copyright holder: Cunze et al.
License: This is an open access article distributed under the terms of the Creative Commons Attribution License, which permits unrestricted use, distribution, reproduction and adaptation in any medium and for any purpose provided that it is properly attributed. For attribution, the original author(s), title, publication source (PeerJ) and either DOI or URL of the article must be cited.
License URL: https://creativecommons.org/licenses/by/4.0/

Keywords: Hantavirus, Puumala virus, Rodent-associated infections, Spatio-temporal patterns

Funding: The authors received no funding for this work.

==============================
Background

Worldwide, the number of recorded human hantavirus infections as well as the number of affected countries is on the rise. In Europe, most human hantavirus infections are caused by the Puumala virus (PUUV), with bank voles (Myodes glareolus) as reservoir hosts. Generally, infection outbreaks have been related to environmental conditions, particularly climatic conditions, food supply for the reservoir species and land use. However, although attempts have been made, the insufficient availability of environmental data is often hampering accurate temporal and spatially explicit models of human hantavirus infections.

Methods

In the present study, dynamics of human PUUV infections between 2001 and 2015 were explored using ArcGIS in order to identify spatio-temporal patterns.

Results

Percentage cover of forest area was identified as an important factor for the spatial pattern, whereas beech mast was found explaining temporal patterns of human PUUV infections in Germany. High numbers of infections were recorded in 2007, 2010 and 2012 and areas with highest records were located in Baden-Wuerttemberg (southwest Germany) and North Rhine-Westphalia (western Germany).

Conclusion

More reliable data on reservoir host distribution, pathogen verification as well as an increased awareness of physicians are some of the factors that should improve future human infection risk assessments in Germany.

Introduction

Hantaviruses (family Hantaviridae) are tri-segmented, negative-stranded enveloped RNA viruses consisting of at least 23 recognized species (Vaheri et al., 2013; Plyusnin & Sironen, 2014). According to the ninth report of the International Committee on Taxonomy of Viruses (ICTV) another 30 species potentially belong to this family. Hantaviruses can be found in different regions worldwide with distributions clearly linked to the distribution of their hosts (Vapalahti et al., 2003). Since the first isolation in 1978 (Lee, Lee & Johnson, 1978), Hantaviruses (named after the river Hantaan in Korea where the first detection of human infections was made) turned into an important subject for researchers and the public, since they constitute an increasing threat to humans with the number of infections globally rising and new viruses being described continuously (Reusken & Heyman, 2013; Lee, Vaheri & Schmaljohn, 2014; Carver et al., 2015). Clinical symptoms and severity of human infections generally depend on the Hantavirus species (Krautkrämer & Zeier, 2014). Two syndromes are known so far: the haemorrhagic fever with renal syndrome (HFRS) and Hantavirus cardiopulmonary syndrome (HCPS) (Krüger, Ulrich & Lundkvist, 2001). Whereas the latter is only found in North and South America, HFRS occurs in Europe, Asia and Africa (Hooper et al., 2001; Witkowski et al., 2014). The most severe clinical course of HFRS is primarily characterized by acute symptoms such as fever, circulatory collapse with hypotension, hemorrhage, and acute kidney injury (AKI) (Jiang et al., 2016). In Europe, the most frequent infections in humans seem to be caused by the Dobrava-Belgrade orthohantavirus (DOBV) as well as the Puumala orthohantavirus (PUUV), with the latter causing a usually milder or moderate form of HFRS, called nephropathia epidemica (NE) (Heyman et al., 2011; Hofmann et al., 2014; Watson et al., 2014). Although rarely fatal (<0.25%), NE patient might exhibit a broad range of flu-like symptoms such as an acute onset of fever, headache, backpain as well as signs of renal involvement that might necessitate more intensive medical attention including ghaemodialysis (Settergren, 2000). PUUV infections, with bank voles (Myodes glareolus) as reservoir host species (Brummer-Korvenkontio et al., 1980; Klingström et al., 2002), are the most prevalent type of human hantavirus infections in Germany (Schwarz et al., 2009), although the DOBV has also been identified in the eastern part of Germany, where the host species of its genotype Kurkino, the striped field mouse (Apodemus agrarius), occurs (Schlegel et al., 2009; Klempa et al., 2013).

Spatial patterns of PUUV infections have been linked to the geographical distribution of the reservoir host (Settergren, 2000; Schwarz et al., 2009; Watson et al., 2014). Thus, it has been suggested that the number of recorded human PUUV infections is directly related to the population density of bank voles, i.e., the higher the abundance of bank voles, the higher the number of infected bank vole individuals (due to the higher chance of transmission between individuals) (Vaheri et al., 2013; Reil et al., 2015), consequently increasing the chances of contact between infected bank voles and humans. In Germany, the heterogeneous spatial distribution of human PUUV infections has been explained by an association of PUUV with the Western evolutionary lineage of the bank vole (Drewes et al., 2017a). Higher abundance of the reservoir host can be largely explained by temporal dynamics of environmental conditions affecting the presence of food resources as well as suitable habitats. Bank voles can adapt to a large range of climatic conditions as can be derived from their vast distributional range covering a broad spectrum of different climatic conditions (cf. Vaheri et al., 2013). Climatic conditions are, however, only indirectly linked to bank vole abundances, i.e., mostly through weather effects on food supply (Imholt et al., 2015). Bank voles are mainly granivorous and profit from high seed production during autumn in mast years (i.e., years with intensive fruiting of beech and oak trees (Fagus sylvatica, Quercus robur and Q. petraea). The high food supply of seeds, mainly beechnuts, favours winter survival of voles and leads to early breeding in spring e(Eccard & Herde, 2013), which in turn results in high number of offspring in early summer ultimately giving rise to PUUV-associated NE outbreaks in humans (Klempa, 2009; Imholt et al., 2015).

The driver of human PUUV infection risk is the likelihood of contact with the virus. The transmission of the PUUV is assumed to be affected by climatic conditions; milder winters and wetter summers are supposed to favour the survival of the virus outside the host species and thus, favour the risk of infection (Heyman et al., 2012). The main pathway of PUUV transmission from bank voles to humans is by inhalation of aerosols contaminated with excreta (urine, feces, and saliva) from infected animals (Ulrich et al., 2008; Schwarz et al., 2009). According to Watson et al. (2014), various outdoor activities can be considered a risk factor for infections, e.g., entering or cleaning long-abandoned places such as cabins or attics, and special risk groups comprise: forest and farm workers, muskrat hunters, military staff, campers, dog owners, cat owners and people living near the forest (Zöller et al., 1995; Sane et al., 2014; Watson et al., 2014).

Figure 1 summarizes the most important factors that are assumed to influence the spatial and temporal patterns of human PUUV infections.

Figure 1 Factors that potentially affect the number of recorded PUUV infections.

Although human infections with PUUV in Europe are on the rise, data from comprehensive long-term monitoring of virus distribution in hosts and patients is often not extensive or incomplete (but see Reusken & Heyman, 2013; Drewes et al., 2017a; Drewes et al., 2017b). Better insights into the role of abiotic and biotic factors and the increasing incidence of PUUV infections in Europe are urgently needed. In Germany, PUUV infections are known since the 1980s (Ulrich et al., 2008). Since the notification requirement in Germany in 2001, data on the number and location of human PUUV infections reported are available at a spatial level of German administrative districts (SurvStat@RKI 2.0, https://survstat.rki.de/Content/Query/Create.aspx/).

The aim of the present study was to visualize spatial and temporal patterns of human infections for the period 2001–2015. We focused on the following questions:

(1) Where are hot spots of PUUV infections in Germany? (spatial patterns)

(2) Which years have been affected by especially high numbers of PUUV infections? (temporal patterns)

(3) Are there any seasonal patterns?

(4) What are the driving factors for the spatial and temporal patterns in Germany?

(5) Has there been an increase of PUUV infections over the last 15 years? Overall or in certain regions?

(6) Does the spatial extent of areas strongly affected by PUUV infections expand? What might be reasons for this?

More specifically, we considered the percentage of strong beech fructification (food supply), the percentage of forest and urban area (land cover) as well as precipitation and temperature (climatic conditions) with regard to their influence on temporal and spatial patterns of the recorded number of PUUV infections in Germany.

Materials and Methods

PUUV infections in Germany

Data on PUUV infections, i.e., the number of reported individual cases of human infections was derived from SurvStat@RKI, provided by the Robert Koch-Institute (RKI) (2017), (http://www.rki.de) at a spatial level of German administrative districts (German: “Landkreise”). For PUUV, the spatial, temporal and seasonal patterns were investigated. We considered the number of recorded infections for each year from 2001–2015 (status as of May 2017) and for each of the 402 German administrative districts with a temporal resolution of one calendar week. The geometry of the German administrative districts and information about the number of inhabitants were obtained from the “Geodatenzentrum”, GeoBasis-DE/ BKG (http://www.geodatenzentrum.de).

Climatic conditions

Data on climatic conditions were provided by the German Meteorological Office (Deutscher Wettersienst (DWD), 2017, http://www.dwd.de, ftp://ftp-cdc.dwd.de/pub/CDC/). Temporal patterns of human PUUV infections were contrasted with mean temperature in July (averaged mean of daily air temperature in 2 m height above ground in July), mean precipitation in July and mean temperature and precipitation in summer (i.e., June, July, August). In addition, we considered annual mean temperature and annual precipitation. Based on the raster data of these variables with a spatial resolution of 1 km2 we first calculated averages of the respective climatic variable (2001–2015) for each grid cell and then calculated averages for each of the 402 German administrative districts. The latter was defined as the final spatial resolution of the study, which was set by the spatial scale available for the data on human PUUV infections.

Habitat structure and beech fructification

The presence of suitable habitats as well as food resources for the reservoir host are considered important factors affecting the temporal and spatial pattern of recorded PUUV infections. We thus took data on habitat structure as well as on beech fructification (affecting the availability of food for the bank voles) into account. For habitat structure we considered the percentage cover of forest and urban area per district. These data were derived from the CORINE Land Cover (CLC) data (provided by the European Environment Agency (2006), https://www.eea.europa.eu/data-and-maps/data/clc-2006-raster-4). The CLC is a seamless European land cover vector database which is based on satellite remote sensing images on a scale of 1:100,000 as the primary information source. Land cover information is separated into 44 classes. For each of the 402 German districts we calculated the proportion of forest considering the classes broad-leaved forest (CLC 3.1.1) or mixed forest (CLC 3.1.3), as well as the proportion of urban area of the classes: CLC 1.1.1, 1.1.2, 1.2.1, 1.2.2, 1.2.3, 1.2.4, 1.3.1, 1.3.2, 1.3.3 and 1.4.2 (i.e., urban, fabric, industrial, commercial and transport units, mines, dumps and construction sites, sport and leisure facilities).

For beech mast we exemplarily considered data from the German federal state of Hesse (located in Central Germany). Data on percentage of old beeches with medium or strong fructification in Hesse were taken from a report on the state of the forest in Hesse in 2016 (“Waldzustandsbericht 2016, Hessen”, Hessisches Ministerium für Umwelt, Klimaschutz, Landwirtschaft und Verbraucherschutz, 2016).

Analysis

To analyse the factors impacting the spatial patterns we correlated the number of recorded human PUUV infections with land cover (i.e., the percentage of forest and urban area) and climatic conditions (i.e., annual mean temperature and annual precipitation) for the 402 districts. In addition, we used a generalized linear model (GLM) to identify variables that could be used to predict incidence rates. Four different variables were included in the model: forest area, urban area, annual mean temperature average over 2001 to 2015 (AMT), annual precipitation average over 2001 to 2015 (AP) and their two-way interactions. Data of percentage cover of forest and urban area was arcsine-transformed and the model used a Gaussian error distribution and an identical link function.

To estimate whether a tendency of spatial expansion of the area strongly affected by PUUV infections exists, we carried out a linear regression over the number of recorded PUUV infections for the years 2001 to 2015 in each district.

Mast years are linked to an increased bank vole population density, which in turn likely increases the numbers of PUUV infections. Drought in early summer seems to be a strong predictor for intensive fruiting of beech in the subsequent year (Piovesan & Adams, 2001). According to Overgaard, Gemmel & Karlsson (2007), temperature in July can also be used as a predictor for beech mast in the next year. The likelihood of a strong fructification after a dry and warm summer was found to be even higher when a moist and cool summer preceded the year before the drought (Piovesan & Adams, 2001). Allowing for these time lags, we correlated the number of recorded PUUV infections of the respective year with beech fructification of the previous year, which in terms was related to the climatic conditions in the years before; mean temperature in July and mean precipitation in July two years before), and mean temperature and precipitation in summer (June, July, August) three years before).

Analysis (e.g., correlation analysis, GLM) was carried out using R (version 3.2.1: R Core Team, 2016). ESRI ArcGIS (version 10.3; ESRI, Redlands, CA, USA) was used for maps as well as for calculating the percentages of forest and urban area. A map outlining the affected regions in Germany is given in Fig. 2.

Results

Spatial patterns of infections were similar between the years 2001 and 2015 with high numbers of recorded PUUV infection in the states of Baden-Wuerttemberg in southwestern Germany (especially in the state capital Stuttgart and the surrounding districts) as well as parts of North Rhine-Westphalia (western Germany) and eastern Bavaria (southern Germany, Figs. 2 and 3). North-eastern Germany was less affected. Within the considered time period of 15 years, there were no human PUUV infection cases in 101 of the 402 districts of Germany (i.e., about one quarter). The maximum number within the considered time period was found in the district “Reutlingen” in the state of Baden-Wuerttemberg (see Fig. 2 for location) with a total number of 573 recorded PUUV infections, of which 188 PUUV infections were recorded in 2007.

Figure 2 Sum of the recorded PUUV infections between 2001 and 2015.

The eight districts that show a significant positive trend between 2001 and 2015 are labelled and hatched. The figure based on data provided by the Robert Koch Institute and the GeoBasis-DE/BKI, and was built using ESRI ArcGIS 10.3 (ESRI, Redlands, CA, USA).

Based on the existing data, there is currently no indication for a clear expansion of the area strongly affected by PUUV infections at the country level. However, in eight out of the 402 districts a significant positive trend, i.e., significant increase of recorded infections from 2001–2015, could be demonstrated (Fig. S1). Five of these eight districts show very low numbers of recorded PUUV infections (up to three per year). The other three districts that show a significant positive trend are situated near the regional hotspot in Baden-Wuerttemberg (Ravensburg), and in North Rhine-Westphalia (Oberhausen and Wesel, Fig. 2).

Figure 3 Incidence of PUUV infections: number of recorded PUUV infections per 100,000 inhabitants in Germany.

Note that most recent data for the year 2016 (P) is shown in this figure for completeness only, but was not included in any other analysis. Figure based on data provided by the Robert Koch Institute and the GeoBasis-DE/BKI, and was built using ESRI ArcGIS 10.3 (ESRI, Redlands, CA, USA).

Although the pattern of recorded infections over the 15 considered years is similar, a strong temporal variation between single years can be observed (Fig. 4). The years 2007, 2010 and 2012 stand out due to their high numbers of recorded PUUV infections, especially within the three hotspot regions: Baden-Wuerttemberg, North Rhine-Westphalia and eastern Bavaria (Figs. 2 and 3). There is no significant trend (p = 0.27) in the annual number of recorded PUUV infections in Germany between 2001 and 2015 (Fig. 4, bottom).

Figure 4 Temporal patterns of the number of recorded PUUV infections and related factors.

(A) Precipitation and temperature in summer (i.e., June, July and August); (B) Precipitation and temperature in July; (C) Percentage of old beech trees with medium or strong fructification in Hesse, Germany; (D) Number of recorded PUUV infections between 2001 to 2016 (2017, data as from 14.11.2017) in Germany. Note the different time lags (also described in Material and Methods). The figure based on data provided by the Robert Koch Institute, the GeoBasis-DE/BKI, the Deutscher Wetterdienst, the Hessisches Ministerium für Umwelt, Klimaschutz, Landwirtschaft und Verbraucherschutz, and was built using R 3.4.2 (R Core Team, 2016).

With regard to the temporal pattern we displayed the relation between the number of recorded PUUV infections (from 2001 to 2015) and the percentage of old beeches with medium or strong and the percentage of old beeches with medium or strong fructification in the German federal state Hesse previous to that year, but also the climatic conditions during previous summers two and three years before (Fig. 4). Significant correlations occurred between the number of PUUV infections and beech mast of the previous year in Hesse (r(s) = 0.69, p < 0.01) and the number of PUUV infections and precipitation in July two years earlier in Germany (r(s) =  − 0.003, p < 0.01. Each year with a high number of recorded PUUV infections was preceded by a mast year but not vice versa (Fig. 4).

There was a clear seasonal pattern especially in years with a high number of recorded PUUV infections, with low values during winter and a clear increase in spring (from the end of April) which reached a maximum in early summer (∼27th calendar week) (Fig. 5). The number of recorded PUUV infections decreased in late summer and autumn. This seasonal pattern was observed when considering the mean of all years between 2001 and 2015 (black line in Fig. 5) and was more pronounced when considering only the years with a high number of PUUV infections (red line in Fig. 5).

Figure 5 Seasonal pattern of recorded human PUUV infections in Germany.

In black: mean average over the years 2001 to 2015 and (in grey: mean average ± standard error), in red: mean average over the three years with a high number of recorded PUUV infections (2007, 2010 and 2012) (in light red: mean average ± standard error), in blue: mean average over the 11 years with a low number of recorded PUUV infections (2001 to 2006, 2008, 2009, 2011, 2013 and 2014) (in light blue: mean average ± standard error). The figure based on data provided by the Robert Koch Institute, and was built using R 3.4.2 (R Core Team, 2016).

Based on current knowledge on factors likely affecting the spatial pattern of the number of recorded PUUV infections in Germany and with respect to data availability, we focused on four environmental factors: the percentage of forest area and urban area regarding habitat structure and annual mean temperature and annual precipitation as climatic variables. The percentage of forest area and the number of recorded PUUV infections was positively correlated (Spearman correlation coefficient r(s) = 0.36, p < 0.001, Fig. S2), which was also confirmed by the GLM with significant interactions of percentage of forest area and annual mean temperature or annual precipitation (Table 1).

Table 1 GLM for the PUUV incidences with four variable.

GLM for the incidences with four variables (forest — arcsine transformed percentage of forest area, urban — arcsine transformed percentage of urban area, AMT — annual mean temperature average over 2001 to 2015, AP — annual precipitation average over 2001 to 2015) and their two-way interactions (Gaussian error distribution). GLM is based on data provided by the Robert Koch Institute, the Deutscher Wetterdienst, the European Environment Agency. Analysis was performed using R 3.4.2 (R Core Team, 2016).

Coefficients:	Estimate	Std. Error	t value	Pr(>|t|)		
(Intercept)	4.12E–04	6.95E–04	0.592	0.554		
Forest	9.59E–03	2.21E–03	4.341	1.81E–05	***	
Urban	−2.95E–04	1.43E–03	−0.207	0.8359		
AMT	−7.10E–06	7.38E–06	−0.961	0.337		
AP	−9.15E–07	5.96E–07	−1.535	0.1257		
Forest:urban	1.38E–03	1.11E–03	1.242	0.2149		
Forest:AMT	−8.21E–05	2.01E–05	−4.074	5.61E–05	***	
Forest:AP	−1.59E–06	7.92E–07	−2.012	0.0449	*	
Urban:AMT	8.69E–06	1.24E–05	0.704	0.4819		
Urban:AP	−8.75E–07	7.00E–07	−1.25	0.2119		
AMT:AP	1.37E–08	7.25E–09	1.884	0.0602	.	
Notes.

Significance codes:

*** 0.001.

* 0.05.

. 0.1.

Discussion

Spatial patterns

Our analyses revealed similar spatial patterns over the considered 15 years and confirmed several hotspots of PUUV infections in Germany: in the federal states of Baden-Wuerttemberg, adjacent areas in Bavaria, the north of North Rhine-Westphalia and the neighbouring part of Lower Saxony (see Weber de Melo et al., 2015). Except for few locations where a significant rise in the number of recorded PUUV infections occurred, no significant positive trend or a clear expansion of the area strongly affected by PUUV infections was demonstrated at the country level, which corroborates results of previous studies (see e.g., Faber et al., 2013; Drewes et al., 2017a). Some of the locations where a rise in infection numbers was noted (i.e., Ravensburg in Baden-Wuerttemberg and Oberhausen and Wesel in North Rhine-Westphalia, cf. Fig. 2 for location of the mentioned districts) are adjacent to already known hot spots of human PUUV infections and may indicate an expansion of the area affected.

Identification of the drivers of PUUV outbreaks is of major importance as it provides the opportunity to predict future outbreaks. For the spatial patterns, forest cover is supposed to be the most important factor, because bank voles as host reservoir species are known to be forest-dwelling. Indeed, we showed that forest cover overlaps with hot spot regions of infections in Germany and detected a highly significant correlation between the percentage of area covered with forest and the number of recorded PUUV infections. Higher risk for human PUUV infections is thus associated with forest rich regions. This association confirms results of previous studies from other European countries; the most plausible assumption and widely suggested risk factor for the transmission of the virus to humans has been their exposure within forested areas through, e.g., working in the forest, living nearby, cleaning utility rooms, visiting forest shelters or using wood for heating and building (see Heyman et al., 2012; Reil et al., 2015). This is also supported by results from the Netherlands where PUUV infections are known since 2008 (Sane et al., 2014). In the Netherlands, numbers of recorded PUUV infections have been generally low (Sane et al., 2014), potentially explained by the relatively low percentage of forest cover in the country. Despite this, peaks have also been recorded for 2010 and 2012 in the Netherlands (Sane et al., 2014), which coincided with two important outbreaks in Germany. Generally, the spatial pattern of the occurrence of infected bank voles is expected to be strongly driven by spatial autocorrelation due to bank voles infecting each other. Data on population dynamics of bank voles and their PUUV seroprevalence would therefore improve the capacity of further testing this relationship (but see discussion on data quality further below). Another strategy would be the establishment of a long-term monitoring of rodents as recently suggested by Jacob et al. (2014).

Temporal patterns

Mast years are commonly considered as main drivers of bank vole abundance in Central Europe (for Germany see Reil et al., 2015). Our results are generally in accordance with this assumption. Strong fluctuations among years occur with PUUV outbreaks in 2007, 2010 and 2012 preceded by a year with a high percentage of fructification of older beech trees (in Hesse). This mechanism seems plausible and has been suggested before: The food supply due to the high seed production in the previous autumn is assumed to favour winter survival of rodents and is thus associated with early breeding in spring, resulting in high densities in early summer, which then gives rise to PUUV outbreaks (Klempa, 2009).

As a consequence of climate change, PUUV outbreaks might become more frequent in the future (Imholt et al., 2015). The three outbreaks of 2007, 2010 and 2012 occurring in rather quick succession in Germany have been considered a possible sign for this climate change induced shortening of outbreak intervals (see also pattern in Reil et al. (2015) and Imholt et al. (2016)). The expected shortening of intervals between years with high numbers of human PUUV infections is due to the shortening of the time intervals between mast years of oak and beech trees in Europe (Overgaard, Gemmel & Karlsson, 2007), which provides optimal food conditions for bank voles in Europe more frequently. Despite the possible positive influence of climate change on the frequency of mast years, the occurrence of mast years remains tied to the presence of nutrient reserves and can thus hardly occur for several consecutive years.

Figure 6 Spatial pattern of four environmental factors that are supposed to affect the spatial pattern of recorded PUUV infections (cf. Fig. 2).

(A) Annual mean temperature (°C); (B) Annual precipitation (mm); (C) Percentage of forest area (%); (D) Percentage of urban area (%). The figure based on data provided by the Deutscher Wetterdienst, the European Environment Agency, and was built using Esri ArcGIS 10.3 (ESRI, Redlands, CA, USA).

However, our results show that not every mast year is followed by a year with a high number of recorded PUUV infections. The years 2006, 2009 and 2011 have been mast years in Germany, followed by years with a high number (>1,000 recorded cases) of recorded human PUUV infections. In contrast, the years 2002, 2004 and 2014 have also been mast years at least for beech in Hesse, but the number of recorded PUUV infections in Germany was only slightly higher than average in the following years (2003, 2005 and 2015) and not comparable to those in the years 2007, 2010 and 2012. In the first few years after 2001, the year of the introducing the reporting obligation, not every case of PUUV might have become reported, which may partially explain lower numbers than expected. In 2005, the fifth highest number of PUUV infections (within years of 2001 to 2016) was observed, with a slightly different spatial pattern. A disease hotspot was present in North Rhine-Westphalia but not in Baden-Wuerttemberg. The higher number of PUUV infections in North Rhine-Westphalia corresponds to the observed higher number of PUUV infections in the neighbouring regions in Belgium (Mailles et al., 2005). In addition, there is a strong spatial variation of the strength of beech mast. Beech mast is supposed to be largely regulated by climate with greater temporal than spatial dynamics. Reil et al. (2015) displayed spatial variation of beech fructification among federal States in Germany for the years 2001 to 2012, which were, however, disregarded in this study as we chose to only exemplarily consider data from Hesse. The year 2014 was considered a Germany-wide mast year, which led to the expectation of a PUUV outbreak in 2015 (Reil et al., 2016). In fact, 542 PUUV infections were reported in Germany in 2015. Although this is the fourth highest number of recorded PUUV infections since the introduction of the reporting obligation in 2001 in Germany, the number was clearly lower in comparison to the PUUV outbreaks in 2007, 2010 and 2012 with 1654, 1893 and 2476 reported cases.

Seasonal patterns

We found a clear seasonal pattern of PUUV infections in years of disease outbreak with its maximum in early summer as previously shown by Krautkrämer, Krüger & Zeier (2012). This seasonal pattern slightly differs from the seasonal pattern in Northern Europe with two seasonal peaks per annum (Vaheri et al., 2013). In Northern Europe a minor peak usually occurs in summer (perhaps related to summer vacation with many people staying outdoors) and a second stronger peak in winter (November–February) (Vaheri et al., 2013). Bank voles undergo 3–4 litters per year, with a peak in abundance in late autumn and winter (Vaheri et al., 2013). The second seasonal peak might thus be ascribed to the high abundance of bank voles during this period. In addition, bank voles tend to frequent the proximity of human dwellings when conditions are adverse, leading to increasing contact with humans and hence infection risk (Olsson et al., 2009; Muyangwa et al., 2015). It was hypothesized that bank voles might move closer to human housings in very cold winters also in Germany, where winter peaks of PUUV infection cases have been observed in some previous years (Faber et al., 2010; Heyman et al., 2012).

Data quality

Apart from the factors used in this study, bank vole abundance and certainly seroprevalence are crucial factors to be taken into account for predictions of rodent-associated human infections. Both factors were recently used in a study by Drewes et al. (2017b), focusing on south-western Germany. Another problem to consider is the current methodology and practice of virus identification. Commonly used methodologies can fail to correctly distinguish antibodies of PUUV and DOBV. In addition, also Tula orthohantavirus can trigger human infections, but cannot be easily identified with standard serological tests but only by neutralization and/or nucleic acid detection and subsequent sequencing (R Ulrich, pers. comm., 2017). Thus, when using data on the respective virus infections, potential identification errors should be taken into account and addressed. More recently, virus shedding (Voutilainen et al., 2015) has been proposed as an alternative method. Another data bias might be introduced through the reporting system; reported infections with rodent-associated pathogens usually refer to the patient’s place of residence, which might be different from the place of infection. As shown here but also recently by Drewes et al. (2017a), north-eastern Germany is less affected. In fact, cases detected in Dessau-Roßlau and Northeim might likely be an example of such mismatch between the location of virus transmission and infection reporting. Furthermore, in the case of a PUUV infection, only 5–10% of the patients display clinical symptoms (Heyman et al., 2009). If mild and strong progressions are recognized as such, a correct analysis of PUUV infection trends might still possible. However, there may be a lack of awareness of some doctors for the disease. Thus, the true number of infections is expected to be much higher than recorded even if awareness of physicians, access to diagnostic tests and reporting mechanisms have been improved in recent years (Watson et al., 2014).

Conclusion

Human PUUV infection hotspots are located in the federal states of Baden-Wuerttemberg and adjacent areas in Bavaria, and the north of North Rhine-Westphalia and the adjacent part of Lower Saxony. There have been strong fluctuations in the total number of recorded PUUV infections among years with high numbers in 2007, 2010 and 2012. Different environmental factors have been discussed and used to better understand and predict the number of human PUUV infections. Interactions between these factors add to the complexity of the situation and may provide the opportunity to further improve models to predict epidemics (Heyman et al., 2012). Reliable long-term, replicated, high-resolution data on population densities of bank voles and other rodents as well as on seroprevalences in host populations could improve the understanding of the ecology of rodent-associated diseases and resulting predictions (e.g., Jacob et al., 2014; Reil et al., 2015; Khalil et al., 2017). Generally, the spatial pattern of infected bank voles is expected to be strongly driven by spatial autocorrelation due to the presence of PUUV in certain regions in Germany and bank voles infecting each other. A higher risk of human PUUV infections can be assumed (i) in areas with a high percentage of forest area (spatial pattern), (ii) in years following a mast year (temporal pattern) and (iii) in early summer (seasonal pattern). Due to ongoing climate change human PUUV infections may gain increased significance in Germany and better risk assessments to predict disease outbreaks are needed.

Supplemental Information

Figure S1 Scatter plot of recorded PUUV infections between 2001 and 2015 for the eight districts that show a significant positive trend between 2001 and 2015 with trend line

Click here for additional data file.

Figure S2 Scatterplots for the number of recorded PUUV infections per 100,000 inhabitants and four environmental factors that are supposed to be relevant

Spearman correlation coefficients (r) between the number of recorded PUUV infections per 100,000 inhabitants and the percentage of forest area as well as the percentage of urban area (derived from the CORINE landcover data) and the annual mean temperature (mean over 2001–2015) as well as annual precipitation (mean over 2001–2015) *** very highly significant (p < 0.001).

Click here for additional data file.

We thank Rainer Ulrich and Jens Jacobs who helped with valuable comments on earlier drafts of the manuscript.

Additional Information and Declarations

Competing Interests

Author Contributions

Data Availability

The authors declare there are no competing interests.

Sarah Cunze and Judith Kochmann analyzed the data, contributed reagents/materials/analysis tools, wrote the paper, prepared figures and/or tables, reviewed drafts of the paper.

Thomas Kuhn, Raphael Frank and Sven Klimpel analyzed the data, contributed reagents/materials/analysis tools, wrote the paper, reviewed drafts of the paper.

Dorian D. Dörge prepared figures and/or tables, reviewed drafts of the paper.

The following information was supplied regarding data availability:

We have not generated any data ourselves. Our analysis is based on freely available data. The references are given in the manuscript. Specifically we used data on the number of recorded PUUV infections provided by the Robert Koch Institute (Robert Koch-Institute (RKI) SurvStat@RKI 2.0, https://survstat.rki.de, data as from 17. May 2017) in Figs. 2–5, and Table 1; data on climatic conditions provided by the “Deutscher Wetterdienst” (Deutscher Wettersienst (DWD) http://www.dwd.de, ftp://ftp-cdc.dwd.de/pub/CDC/, data as from 14. November 2017) in Fig. 4, and Table 1; the geometry of German districts and data on population numbers provided by the GeoBasis-DE/BKG Geodatenzentrum (2016) (GeoBasis-DE/BKG, Geodatenzentrum http://www.geodatenzentrum.de, data as from 28. January 2016 (modified)) in Figs. 2–6, and Table 1, as well as data on land cover, i.e., the CORINE land cover data provided by the European Environment Agency (European Environment Agency (2006), CORINE land cover data https://www.eea.europa.eu/data-and-maps/data/clc-2006-raster-4, data as from 5. December 2016) in Fig. 6, and Table 1.

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
