# Peer review of "Spatial and temporal patterns of human Puumala virus (PUUV) infections in Germany"

_PeerJ, doi:10.7717/peerj.4255_

## Round 0.1 · original submission · Major Revisions

While this is an interesting and well-presented study, there are some substantial revisions required. The main issue with the manuscript are the absence of any reference to the any of the other 7 rodent transmitted pathogens in the abstract and introduction. These pathogens appear first in the materials and methods, which is rather confusing to the reader. The paper stands on it's own without the reference to these other pathogens, which would justify another manuscript.

I think it is also important to adjust the content of the introduction to focus more on the relevance of PUUV infections to human health.

Finally, as someone who is not familiar with geography of Germany, I found the references to the different regions rather difficult to follow. As suggested by the reviewer, please review the way you refer to these, perhaps by using NESW teminology to make it easier to visualise for the reader. Alternatively, provide an annotated map for the reader to refer to showing the names of the regions.

·

Basic reporting

1. The title is a bit misleading since it’s not clear if it’s human or bank vole infections with PUUV the authors refer to. Also, PUUV should be given directly after Puumala virus. Therefore, I suggest to change the title to either “Spatial and temporal patterns of human Puumala virus (PUUV) infections in Germany” or to avoid any confusion: “Spatial and temporal patterns of nephropathia epidemica (NE) incidence in Germany”.

2. The manuscript (as also indicated by the title) deals with PUUV. Why are other pathogens included in the manuscript? This inclusion is not motivated in the introduction and does not get sufficient attention in the analyses/results and discussion. In my opinion, the authors should exclude the other pathogens completely. They can be the topic of another and completely different manuscript. Focusing on PUUV, the manuscript is sufficiently interesting. For PUUV, the bank vole is the only reservoir host. For the other diseases/pathogens, other mammals are involved and even vectors. This aspect (apart from many other issues) requires much more attention than given right now. Also, other mechanisms are probably involved in disease transmission to humans.

3. L. 16 but also l. 54-55. To my knowledge, the bank vole is the only competent reservoir host, i.e. not only the main host as stated by the authors. References: Klingström, J. et al. Rodent host specificity of European hantaviruses: Evidence of Puumala virus interspecific spillover. J. Med. Virol. 68, 581–588 (2002). Brummer-Korvenkontio, M. et al. Nephropathia Epidemica: Detection of Antigen in Bank Voles and Serologic Diagnosis of Human
Infection. J. Infect. Dis. 141, 131–134 (1980).

4. L. 30. There is no need to include the German word “Meldepflicht”. In English, such a disease is called a notifiable disease and the sentence should be changed accordingly. The authors actually use the term in the manuscript (l. 108) and should do so also in the abstract.

5. The authors refer in the introduction to the ecology of bank voles and mention transmission risk among bank voles, but don’t explain how humans contract NE. Where and when is PUUV transmitted to humans? What is the main contact zone? This is shown in Figure 1, but should also be explained in the text. In fact, the authors need to discuss different aspects of the contact zone in much more detail. In the discussion, it’s only mentioned in one section (l. 319-322). It’s interesting that the authors found a significant relationship with forest cover in their analyses. However, this relationship just reflects potential hotspots of the bank vole as a forest dwelling species. HOW do people get infected? In the forest? In the discussion, the authors refer to outdoor activities in Northern Europe (l. 314-315). But it’s hardly walking in the woods that is a human risk behaviour. Also with reference to Vaheri et al. 2013, the authors should be more specific in terms of which activities that increase transmission risk in northern Europe. More important though: What are the high-risk activities in Germany? Are they the same as in northern Europe? How does this fit with the temporal peaks of human PUUV infection in northern Europe and Germany. It’s this kind of discussion that would be really interesting and improve the manuscript.

6. L. 98. What kind of spread do the authors mean?

7. L. 110. Explain TBE and not only refer to the German term. But see my general comment to delete other pathogens/diseases.

Experimental design

1. L. 114. PUUV is a notifiable disease in Germany since 2001 (l. 86). The authors use human PUUV data since 2001. There is hence a risk that at least in 2001 human PUUV cases are not as frequently reported (due to unawareness). Have the authors considered this potential bias in their analysis? It’s mentioned in the discussion, l. 293-295, but what are the implications for their study? Should the authors for example exclude data for the first and even second year from their analyses?

2. L. 121. Mention that ArcGIS is a software for analysis of GIS-data. ArcGIS is mentioned at three places in the methods. It should be sufficient to mention once that you did all spatial analyses with this software.

3. L. 152. What is “Hesse”? A state? Clarify.

4. General aspect in Methods. For some datasets, the authors give the spatial resolution of their data (forest data). But what is the spatial resolution of all other spatial data that were used and at what spatial resolution were the analyses finally performed?

5. L. 131 and 156. The authors performed generalized linear modelling. More details on these analyses need to be given. Any data transformations? Which distribution was used and which link function was used? Table S1 gives some details, but not all. All details need to be given in the Methods.

Validity of the findings

1. Figure 2. For readers, not familiar with German geography, mentioning the different states does not help to understand the maps. Either the authors should refer to geography in terms of NSEW or the authors should include a separate map of Germany where they indicate the states they refer to. Since data from 2016 are not included in the analyses, the authors could for example delete the map for 2016 and use this panel for the suggested map, instead. In the legend: What does RKI refer to? I don’t think it’s necessary to give the projection type in the legend.

2. Also with reference to my comment to Fig. 2, the authors mention several regions and cities in the manuscript (Northeim, Dessau-Rosslau). Sorry, but these places don’t mean anything to readers outside Germany. The authors need to find a better way to relate to geographic locations.

3. L. 177. Do the authors really want to refer to Fig. 3? Shouldn’t it be Fig. 2? Otherwise, I don’t understand this sentence.

4. L. 178-179. Isn’t this contradictory to what is mentioned in l. 168-170? If there is an increase in infection numbers in some areas, but not the country level, this should be clarified in the text. How was the trend tested? There seems to be cyclicity in the data. Was that considered?

5. Figure 3. Sorry, but I don’t understand the x-1 . x-3. To increase readability, the authors should use the same time scale for all panels. How is summer defined (which months)? In the lowermost panel, the authors refer to “HV infections”. Do you mean human PUUV infections? Should be explained in the legend. HV infections and PUUV infections are used in parallel. To not confuse the reader, please use only PUUV infections.

6. L. 180-187. The x-1 x-2 aspect is not easy to understand (see also my comment to Fig. 3). If lags are used, this should be mentioned in the methods. Also the value of the lag-phases needs to be motivated.

7. Figure 4. Please change the colour or pattern of the black or blue line. Even with a version printed in colour, it’s not possible to see the difference between the two.

8. Results related to Fig. 5/Table S1. Table S1 includes the actual interesting results, whereas Fig. 5 just illustrates the results with a map. Table S1 should be moved to the result section.

9. Please check citations in the text. Parentheses are sometimes put at strange places (e.g. Reil reference l. 236 and Jacob reference l. 247.

10. L. 284-286. Statement should be supported by references.

11. L. 333-334. Sorry, I don’t understand what the authors mean with this sentence in relation to the previous sentence.

---

## Round 0.2 · accepted · Accept

Thank you for responding to the reviewer comments.